# A Mixed-Reality Tele-Operation Method for High-Level Control of a Legged-Manipulator Robot [note 1]

**DOI:** 10.3390/s22218146

**Published:** 2022-10-24

**Authors:** Christyan Cruz Ulloa, David Domínguez, Jaime Del Cerro, Antonio Barrientos

**Affiliations:** Centro de Automática y Robótica (CSIC-UPM), Universidad Politécnica de Madrid—Consejo Superior de Investigaciones Científicas, 28006 Madrid, Spain

**Keywords:** mixed-reality tele-operation, quadruped robot, ROS, arm manipulator, search and rescue, walking robots, Gazebo

## Abstract

In recent years, legged (quadruped) robots have been subject of technological study and continuous development. These robots have a leading role in applications that require high mobility skills in complex terrain, as is the case of Search and Rescue (SAR). These robots stand out for their ability to adapt to different terrains, overcome obstacles and move within unstructured environments. Most of the implementations recently developed are focused on data collecting with sensors, such as lidar or cameras. This work seeks to integrate a 6DoF arm manipulator to the quadruped robot ARTU-R (A1 Rescue Tasks UPM Robot) by Unitree to perform manipulation tasks in SAR environments. The main contribution of this work is focused on the High-level control of the robotic set (Legged + Manipulator) using Mixed-Reality (MR). An optimization phase of the robotic set workspace has been previously developed in Matlab for the implementation, as well as a simulation phase in Gazebo to verify the dynamic functionality of the set in reconstructed environments. The first and second generation of Hololens glasses have been used and contrasted with a conventional interface to develop the MR control part of the proposed method. Manipulations of first aid equipment have been carried out to evaluate the proposed method. The main results show that the proposed method allows better control of the robotic set than conventional interfaces, improving the operator efficiency in performing robotic handling tasks and increasing confidence in decision-making. On the other hand, Hololens 2 showed a better user experience concerning graphics and latency time.

## 1. Introduction

The development of specialized robots endowed with custom instrumentation to assist and provide information to their operators is possible thanks to the advances in robotics, communication and sensory systems, as well as technological development such as Virtual and Mixed-Reality. Quadrupedal robots have had great prominence in recent years in exploration tasks, over conventional locomotion robots (tracks or wheels) [1]; mainly due to their ability to move through unstructured terrain [2,3].

The importance of quadruped robots as exploration platforms in post-disaster environments has been increasing in recent years, proving their ability in search and rescue (SAR) tasks, for example, in one of the works developed by the authors, focused on detecting victims using thermal images and the ARTU-R (A1 Rescue Task UPM Robot) [4].

Most of the works carried out by using quadruped robots are focused on data capture [5,6,7,8,9,10,11]. Nevertheless, restricting the use of SAR robots to pure data acquisition or monitoring missions is a very conservative ambition. A more challenging goal would be to endow the quadruped with manipulation capability by integrating it with a manipulator on board as sport arm by Boston Dynamic [12]. For this propose, a first phase of workspace optimization has been developed with Matlab and a Gazebo simulations to verify the viability of the prototype from the dynamics point of view.

The movement control of the these set is complex when using conventional interfaces (those based on a single o multiple screens, mouses and keyboard), mainly due to the lack of situational awareness suffered by the operator that relies on a poor environment perception. Some developments have sought to address this problem by providing tele-operation solutions. However, these solutions turn out to be quite invasive [13,14,15].

The main contribution of this work is focused on the implementation of a Tele-operation method for ARTU-R and the six Degrees of freedom (DoF) robotic arm based on a Mixed-Reality (MR) system, where the operator becomes aware of the environment through a virtual model projected on the real work through an ergonomic, untethered self-contained holographic device, the Hololens (by Microsoft). The proposed solution allows commanding movements in the workspace, using his hands and a visual mark placed on the robot end-effector.

Experiments have been carried out to improve the efficiency in medical equipment delivery tasks. One of the main advantages of MR implemented system is to increase the operators confidence due to the “first person view” that allows to identify the type of object and perhaps its fragility, deformability, etc.

The validation tests have been carried out in post-disaster scenarios reconstructed according to the specifications of the NIST (National Institute of Standards and Technology).

A first version of this work has been presented as part of CLAWAR-2022 [16], which has been extended in detail and complemented with an outdoor testing phase. Moreover, a comparison section among conventional interfaces (Hololens 1st and 2nd generation) has been included, in order to perform an analysis of the state-of-the-art systems for this Mixed-Reality study.

This work is structured as follows. Section 2 shows the most relevant works related to manipulation platforms based on legged robots and their commanding solutions; Section 3 details the materials and methods. Section 4 contains the experiments and results. Finally, Section 5 presents the main findings.

## 2. Related Works

### 2.1. Legged-Manipulator Robots

Quadruped robots have been an object of study within the field of robotics in the last decade. In these studies, explorations and balance studies have been carried out [17], the behaviour of trajectory planning, gait modes and commercial platforms for example the ANYmal [18].

The environments that the robots need to face up in Search and Rescue missions can be very hostile. Some studies works on an adaptive control to all types of surfaces, such as snow, water or leafy vegetation, making it difficult to perceive the scene [19].

Several implementations have been carried out with quadrupedal robots for search and rescue tasks focused on different objectives such as delivering objects to the required place, picking up objects and placing them to a safer place or just for obtaining data. Some of the most recent developments in this field are RoLoMa and ALMA, which use as development base the ANYmal quadruped robot, equipped with a (${\mathrm{Jaco}}^{2}$) robot (6 DoF) manipulator. Different analyses have been developed, such as dynamic and stability and stability and application tasks as door opening [20,21]. Other studies focuses on the study of generated forces in pick and place task [22].

Different control methods as deep learning have been used to minimize the uncertainty and errors during task executions, as the development in [2], which uses the ALMA robot based on the ANYmal robot. Some studies as the one by P. Mitrano [23] centers on the problem of operating objects with unexpected characteristics.

### 2.2. Legged-Manipulator Robots Tele-Operated by Mixed-Reality

The tele-operation of a robotic set is conceived as a complement to the High-Level control, also analyzed in this work, with references to previous works such as [24].

The tele-operation of a manipulator allows to execute a task such as pick-place in a remote way. In the case that the operator had visual contact with the work environment a mixed reality solution would be feasible whereas a virtual reality one [25] is the appropriate solution when there is no visual information and the feedback has to be artificially recreated [26].

Interfaces with the necessary basic information are developed to represent the work environment of mobile robots. Thus, the manipulation or simply the movement in real-time is better [27]. This environmental information becomes more critical in unstructured environments, where recreation can be more complex [28].

The most recent works developed in this area have conceived tele-operated prototypes using joysticks and VR glasses [13,29,30].

There are other novel methods for high-level control that use gestures or voice. These studies have focused on robotic manipulators such as [31,32,33,34,35,36], which have been considered a reference for the tele-operation of the 6DoF manipulator used in this work.

## 3. Materials and Methods

### 3.1. Materials

The Robotics platform used for this development is a quadruped of the Unitree A1 model ARTU-R (A1 Rescue Task UPM Robot), equipped with a sensory system and a robotic arm described in Table 1. Both the Hololens 1st and 2nd generation glasses have been used for tele-operation with Mixed Reality (Figure 1). The experimental test has been developed at the facilities of the Centre for Automation and Robotics of the Universidad Politécnica de Madrid (UPM).

### 3.2. Robotic Set Design and Modeling

Both the modelling and design of the set have been initially implemented in Matlab. The direct kinematic models have been obtained independently for the manipulator and one leg of the quadruped robot. In both cases, the Denavit-Hartenberg parameters have been calculated and they are shown in the Table 2 and Table 3 as a convenient reference system.

#### 3.2.1. Matlab Application for Kinematics Integration-Visualization

An application has been created in Matlab for joint visualization of the robot and its workspace (Figure 2a). All links are generated and located, as previously revealed. Prismatic shapes are also generated to represent the A1 body, the sensor layout area, and the arm-body mechanical coupling. Finally, the values of the start angles of the legs are set.

The assembly and the workspace graph generated are represented on the same Matlab figure as Figure 2 shows. The application allows all the DoF to be varied (6 for the arm sliders and 3 for each leg sliders). The code that places the robot in a global reference system has been developed for later implementation. This code uses initial values of each joint and initial position and orientation values concerning reference system.

The kinematic model is modified in a loop with as many iterations as there are variations in the joints from this point. Intuitively, an iteration is considered as the movement of a fixed point to the subsequent one, which is also static. In this way, knowing the initial position of the vector, we can modify it successively according to the new support points of the quadruped.

#### 3.2.2. Workspace Definition

The local reference systems of each leg and the robotic arm are allocated in the robot torso, as shown in Figure 2b. The entire assembly reference system has been placed in the centre of the A1 body, below the arm system.

Different parameters such as: the quadruped height and position and the robotic arm position are predominant in determining the optimal workspace. This is created from iterating and saving the values (x,y,z) obtained from slight variations in the 6 degrees of freedom that the arm has.

**Figure 2 sensors-22-08146-f002:**
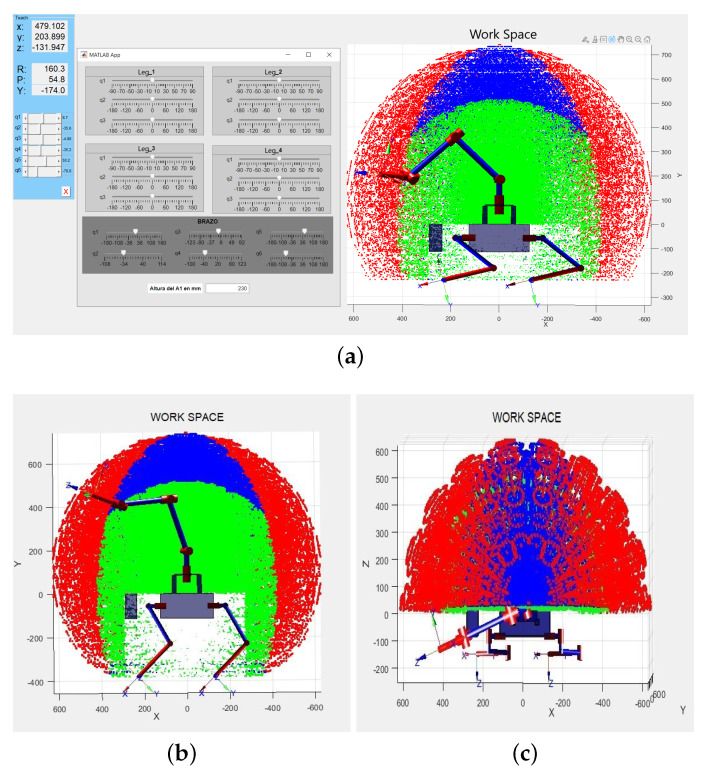
Kinematic Model and Workspace of Robotic Set in Matlab. (**a**) Matlab interface developed for Work Space optimization. The quadruped-manipulator robot is shown in reduced height position. (**b**) Lateral Workspace view of the Model. Reference system is located in the torso center. (**c**) Upper Workspace view of the Model.

Considering Kinematic restrictions of the robotic set, the robot end-effector would be able to reach any point of a spherical, which covers from a radius of 650 mm to 740 mm. It is shown in red colour in Figure 2b,c. Nevertheless, according to the manufacturer recommendation regarding the optimal work zone, it is recommended to work within a 70% or the maximum range; this is shown by the green zone.

There is a high risk of collision with the robot’s head or rear. The different point colours in Figure 2a–c were established to differentiate the manipulation zones that the robot can reach. Therefore, the green area is optimal for manipulation tasks. However, red zones are not considered for manipulation due to the generation of high torques or the manufacturer’s recommendation.

It has been considered that the manipulator has more performance in the lateral areas than in the front and rear, for stability reasons. An ellipsoid with its semi-major axis in the transverse direction of A1 is shown in blue in Figure 2b.

### 3.3. ROS-Gazebo Implementation and Simulation

Different test have been executed in ROS-Gazebo to evaluate the kinematic and dynamic stability of the set as well as the collision free paths for the manipulator arm (Figure 3). Movement simulations of the robotic set have been developed in recreated indoor and outdoor Gazebo environments. One of the main advantages of this simulation is that the arm perception system takes into account the quadruped robot, avoiding collisions (self-collision) during the generation of trajectories close to the robot thanks to the MoveIt and ROS-Control packages.

The simulated environments, recreated according to post disaster scenarios are shown in Figure 3a,b,d,e (indoors), g,h (outdoors). These environments include unstructured soil, small debris, victims, and the robotic set integrated through an assembly carried out by using a xacro macro language. Figure 3c,f,i correspond to collision free trajectories of the respective Gazebo environments. The target configuration of the arm is shown in orange and the current one is shown in black.

### 3.4. Mixed-Reality Tele-Operation System

One of the main problems when controlling this type of complex platform arises from conventional interfaces (remote screens) since they do not provide the operator with an immersive sensation and generate a certain degree of mistrust when carrying out a movement. A system based on Mixed-Reality is proposed for high-level control of the robotic system to address this problem (Figure 4).

The operating scheme of the tele-operation system is shown in Figure 4a. The operator wears Hololens glasses as human-robot-environment interaction. In Hololens, both the physical environment and the virtual robot model are displayed. A blue sphere attached to the manipulator end-effector can be moved through gestures in order to provide high level commands to the robotic set. Thus, the values of the joints obtained in simulation are sent to the real robot collision free planner to execute the trajectory.

Figure 4b describes the block diagram used to control the robotic assembly. The different systems are marked with colour blocks. The referential inputs taken from the virtual model are received from the Hololens. Different goal states are sent to collision-free planners to generate both the path of displacement, as well as the movements of the arm for manipulation.

The variables referring to the quadruped robot position and orientation are given by x,y,θ for both the goal and current positions. The trajectory global planner generates the discretized path $(x_{i\ldots n},y_{i\ldots n}$) with *n* goal points, while the control decision system generates the distance and orientation errors ($e_{ang},e_{dist}$) between the current and goal position. Finally, angular (ω) and linear (*v*) velocities are sent to the robot, until the trajectory is complete.

On the other hand, the robotic arm control is similar, Collision Free Planner takes the current and destination joint positions (($\theta_{1}$, …, $\theta_{6}$)), together with the current state of the quadruped robot. The planner generates the errors of each joint ($e_{\theta_{1}}$, …, $e_{\theta_{1}}$) which are the controller input, finally velocities ($\omega_{1}$, …, $\omega_{6}$) are sent to the robot.

Figure 4c,d shows the user wearing the Hololens 2 and the robotic set in an outdoor scenario. Reference systems representation of the quadruped robot and the tele-operated manipulator arm are shown. Blue line represents the control interaction from the user hand to the robot end-effector.

The Cartesian coordinate system corresponding to the hands has been designed so that both right-handed and left-handed people can use it.

The fingers movements are detected by using the functions of HoloToolkit package in Unity (Manipulation Mode function), which captures the fingers union showing the interaction with the virtual objects (end of the robot). Once the target configuration has been defined, the IK is calculated and then, the joint positions are sent through RosBridge. The MR interface relies on additional buttons that execute predefined commands (home and sleep) and control the gripper actions (open-close).

The strategy used for transporting payload using the implemented robot and the MR system is based on an approach phase, where the robot with the arm in the collected position without load moves, minimizing its gravity centre towards point A. As long as it is not being manipulated, the ideal is to pick up the manipulator and work with a control model that dampens the oscillations generated in the quadruped body since it is the area where it joins the arm.

The user is allowed to command the set at a maximum distance of five meters to pick the object with the gripper. The lateral posture is the most favourable to proceeding to load an object since the quadruped robot is capable of adopting configurations with more stable joints to support the object’s weight.

Once the object has been picked up, the arm acquires a transporting configuration while the quadruped robot moves to the victim surroundings (point B). Finally, the delivering phase is carried out in tele-operated mode.

**Figure 4 sensors-22-08146-f004:**
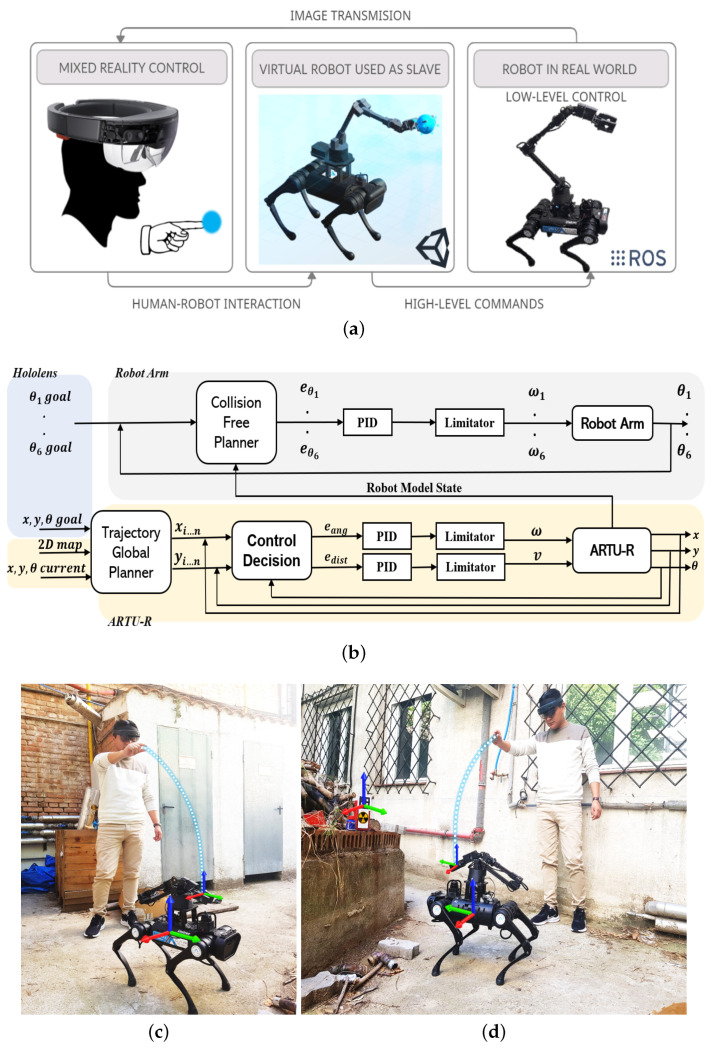
Description of the Mixed-Reality system for tele-operation of the robotic set. (**a**) Systematic scheme of tele-operation of the robot-manipulator through Mixed-Reality. (**b**) Systematic scheme of tele-operation of the robot-manipulator through Mixed-Reality. (**c**) Schematic reference systems representation of the quadruped robot and the tele-operated manipulator arm-Frontal View. (**d**) Schematic reference systems representation of the quadruped robot and the tele-operated manipulator arm-Lateral View.

## 4. Experiments and Results

Several test have been performed to validate the operability of the robotic set through MR (Figure 5, Figure 6 and Figure 7), by using both models of Hololens and conventional interfaces to determine their advantages and limitations. The MR tele-operation system has been evaluated in terms of versatility, immersive experience and user confidence to control of the robotic set (Figure 8). The performance tests are described in Appendix A.

Two types of questions were used, one with multiple alternative responses referring to the type of interface preferred and questions with a quantitative response [0–10] to evaluate the confidence level experienced and the percentage of efficiency with which it has been completed. After completing the tests with both types of interfaces, the participants filled out survey forms, which were tabulated to obtain the graphs and information shown in this section. The task and experiences experienced during the execution and times of task completion have also been measured.

### 4.1. Evaluation of the Mixed-Reality System for Tele-Operation

Two different types of interfaces were used for tests. The first is the MR Tele-operation system, and the second is a conventional one (based on a RVIZ interface); Both have the same purpose, to command movements on the manipulator robot, considering the self-collision. The tests were carried out with 10 participants in the role of robotics set operator, using both interfaces to perform the same task, which consisted moving the end-effector from point A and transporting it to point B, marked with an X on the ground.

Figure 5a–c show the operator’s hand picking the blue ball and performing the movements in space. These Figures show the superimposition of the robots’ virtual model (grey colour) on the real robot (black colour) in such a way that as the blue ball moves, the virtual model calculates the robots’ positions to send them to the real robot controller. A maximum delay of ms has been observed during this first phase of experimentation.

Figure 5d,e show the box diagrams of the data collected, where participants’ experiences show that the implemented system generates a better experience for immersive situations by 15% and increases confidence when executing movements with the robotic arm by 23% compared to conventional interfaces.

### 4.2. Pick and Place Test Using MR System

The next phase of experiments consisted of moving the robot’s end-effector from point A, taking a medical-type object and depositing it at point B (marked with an X on the ground). Objects used for this test were a bottle of alcohol and a first aid kit.

Figure 6a–d shows the steps to deliver an alcohol bottle (150 g weight) from point A to point B. The stage of handling and capturing the object takes 7 s. Its cylindrical shape facilitates the process, but its transport to the destination point takes 12 s more because the operator requires greater precision in the movements to avoid the fall of the object with liquid that can damage the robot.

On the other hand, Figure 6e–h shows the different stages to transport a first aid kit (100 g weight), starting from t=0 s. The object has been reached and picked up in the first 10 s. This first stage takes 3 s more than the previous case, since the rectangular shape of the object requires orienting the robot end-effector. An additional time of 10 s has been required to reach the destination position and drop it.

### 4.3. Victim Assistance Test in Reconstructed Post-Disaster Environments

The third phase of tests has been developed in reconstructed environments (indoors and outdoors), consisting of victims (mannequins and people) and small debris (Figure 7).

In this tests, the task consist in take a first-aid kit from point A to point B (victim). In the first instance, robot approaches and takes the payload (Figure 7a–c,f), places the arm in the transport position and moves through the environment until reaching the victim (Figure 7d,g). Finally, robot extends his arm and drops the package near the victim (Figure 7e,h). The entire process is tele-operated by the robot from the remote control and the arm from the MR system.

### 4.4. Mixed-Reality and Common Interfaces Comparison

The following graphs have been established (Figure 8a,b), based on the operators experiences collected. Which describe the efficiency of executing tasks based on the operators’ complexity and previous experience.

Evolution of the efficiency to execute tasks based on training using both types of interfaces is represented in Figure 8a. At starting point, conventional interfaces are shown to be better since they are intuitive. However, after a series of training sessions (20 tests), MR interface is better than conventional. However, there is a turning point where MR surpasses the conventional (16 training test). This is due to several factors, one of which is the immersive experience generated by using these devices.

On the other hand, Figure 8b shows the efficiency evolution based on the task complexity (which defines if a task is easy to perform or not) in both cases, executions with MR and conventional interface. Both of them shown high efficiency for simple tasks such as moving the robot arm from a point A to a point B. However, for more complex tasks such as object manipulation, the Mixed-Reality system is superior by 21%.

**Figure 8 sensors-22-08146-f008:**
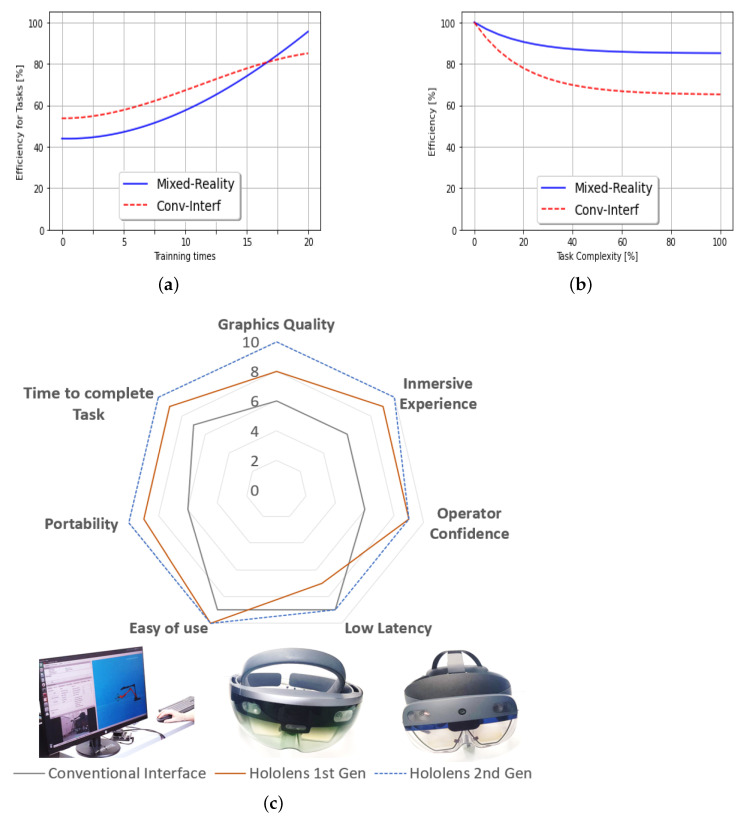
Evaluation metrics to analyze the performance of conventional and MR interfaces. (**a**) Efficiency based on training analysis. (**b**) Task complexly evaluation. (**c**) Parametric comparison of the technological elements for tele-operation the robotic set through conventional interface (screen, mouse, keyboard) and MR interface (Hololens 1st and 2nd generation).

Figure 8c shows a hardware technological interfaces comparison to control the robot set. Different relevant points are analyzed quantitatively to establish comparative metrics based on the experience of the operators (10 people). The following aspects have been analyzed using a radial graph: Immersive experience, Operator confidence, Low latency, Easy of use, Portability, Speed to complete the task and Graphics Quality.

Based on the results obtained, it can be established that depending on an evolutionary line of technological development; each device analyzed has integrated improvements concerning its predecessors. Thus, in ascending order, the hardware prioritized to develop the tele-operation of the robotic set are: conventional interfaces (screen, mouse, keyboard), Hololens 1st and 2nd generation.

The main advantage of the proposed system is the versatility it offers the user, making possible the gestural control of a complex robotic system and providing a more direct interaction of the environment, the robot and the workspace. The portability of the system should also be highlighted since it depends only on the glasses, a laptop as a central communications system, and the robot, which is a great advantage over other proposed methods that use large equipment and, above all, high costs, such as those developed in [13].

#### Advantages compared with State-of-the-Art Work

In this section, a comparison of the main related works that use mixed reality for similar applications is established, highlighting the robustness of the proposed system over the previous ones. Table 4 shows the characteristics of related jobs.

In the first instance, it should be noted that few works related to tele-operation through MR of mobile robots equipped with manipulators. The work presented at [13] is the closest since it uses a robot-legged-manipulator and tele-operation through virtual reality as a platform. However, in contrast to the method proposed, its costly, invasive equipment for the operator and large size are used, making portability difficult.

On the other hand, [37] proposes a development with the KUKA-youBot platform (omnidirectional base and manipulator’s arm), which highlights the functionality of Hololens for data visualization in navigation tasks, but no actions are executed. Gesture command controls the manipulator as in the one proposed by the authors.

In most tele-operation works using MR, the base of the manipulators are anchored to fixed bases which reduce the complexity of tele-operation but limits its use to a restricted work area as in the works developed in [38,39,40].

Some works like [41] focus on setting target positions for the robot and using an off-line phase for planning and executing movements asynchronously.

## 5. Conclusions

Optimizing the work area through kinematic modelling has made it possible to define the lateral areas of the quadruped robot as optimal work areas, maintaining the stability of the robotic assembly, avoiding over-stress in restricted positions and occlusion of the sensory systems of the robot.

The simulation phase in Gazebo before the implementation of the physical assembly has allowed for validating the dynamic stability conditions of the assembly for manipulation tasks and the generation of collision-free trajectories for the manipulator.

Mixed-Reality interfaces for tele-operation have shown better results concerning conventional ones. Some relevant aspects include the operator’s increased confidence for decision-making and the improvement in efficiency to perform complex handling tasks with objects with irregular geometry and weight.

Hololens 2nd generation has shown better results for high-level control than Hololens 1st generation and conventional interfaces, as they improve graphic quality, response times, latency reduction and versatility of use. This technology for tele-operation has a significant advantage since it is portable compared to conventional technologies composed of a screen and keyboard.

Regarding future works, the study of remote operations is proposed, where the operator and the robotic system are in different scenarios (safe place-catastrophic place) connected through the 5G network, as well as the use of auxiliary perception systems that provide the operator with additional information of the decision making environment.

## Figures and Tables

**Figure 1 sensors-22-08146-f001:**
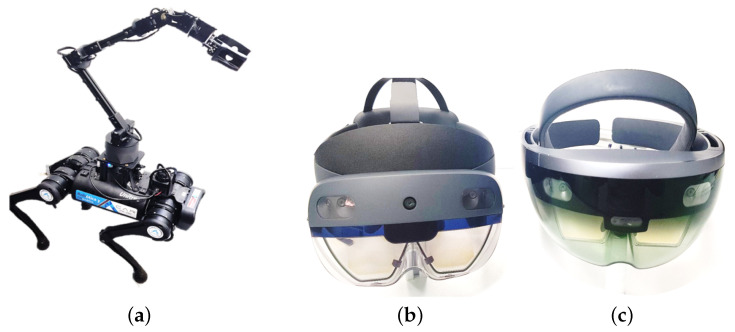
Materials used to validate the proof of concept. (**a**) Robotic set (ARTU-R + 6DoF Robot Arm). (**b**) Hololens Microsoft 2nd Generation. (**c**) Hololens Microsoft 1st Generation.

**Figure 3 sensors-22-08146-f003:**
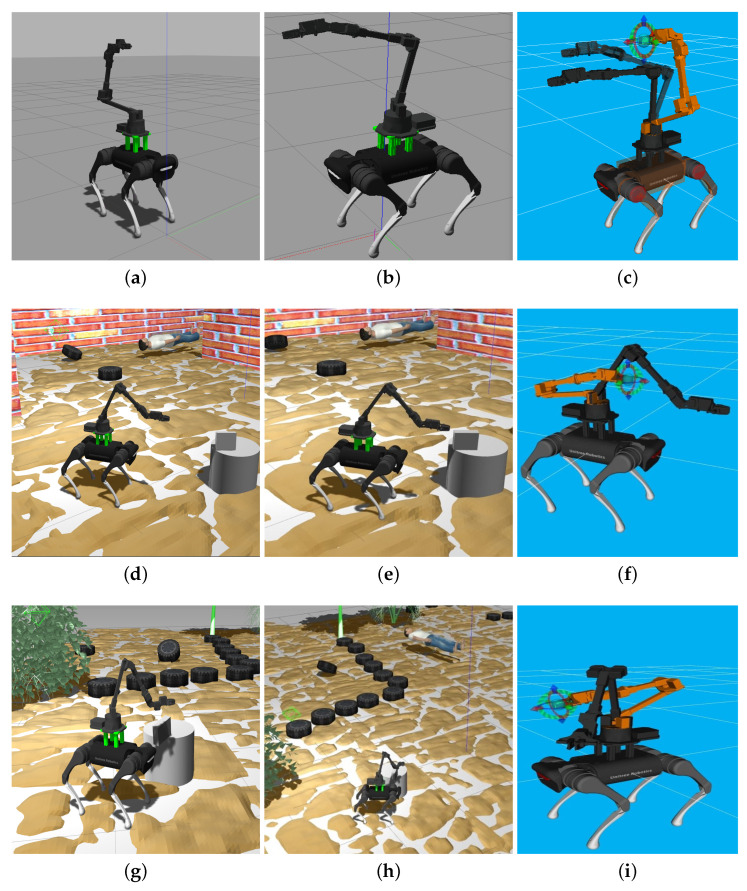
Simulation of the Legged-Manipulator Robot for manipulation tests using Movement Planning (MoveIt) in different Gazebo environments. (**a**) Simulation of the robotic set in Gazebo. (**b**) Simulation of the robotic set in Gazebo. (**c**) Collision free path trajectory planned by Moveit. (**d**) Simulation of Manipulation with the robot set (indoors). (**e**) Simulation of Manipulation with the robot set (indoors). (**f**) Planning of movements by MoveIt. (**g**) Simulation of Manipulation with the robot set in Gazebo (outdoors). (**h**) Manipulation simulation in Gazebo (outdoors). (**i**) Planning of movements by MoveIt.

**Figure 5 sensors-22-08146-f005:**
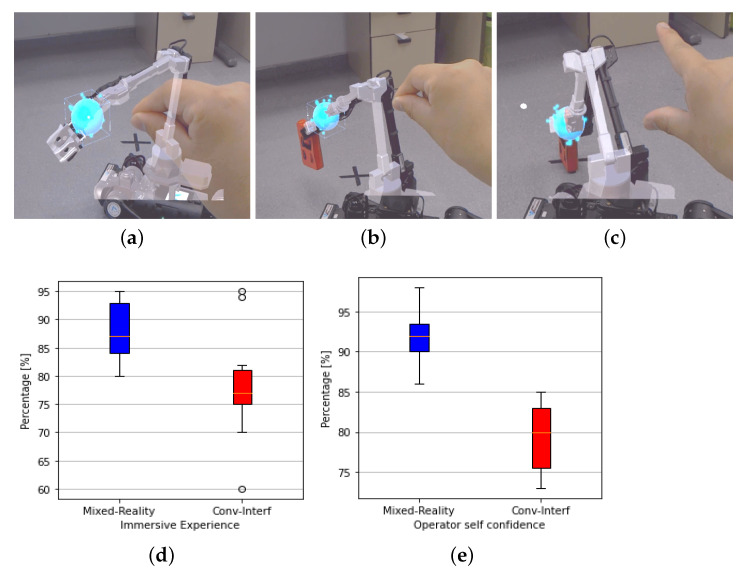
Tests to evaluate the Mixed-Reality efficiency system, the operator holds the robots end (blue ball) and using his gestures, moves it towards the destination (X Mark). (**a**) Tele-operation in home position. (**b**) Tele-operation in 2nd Pose. (**c**) Tele-operation to final position. (**d**) Immersive experience analysis. (**e**) Operator confidence analysis.

**Figure 6 sensors-22-08146-f006:**
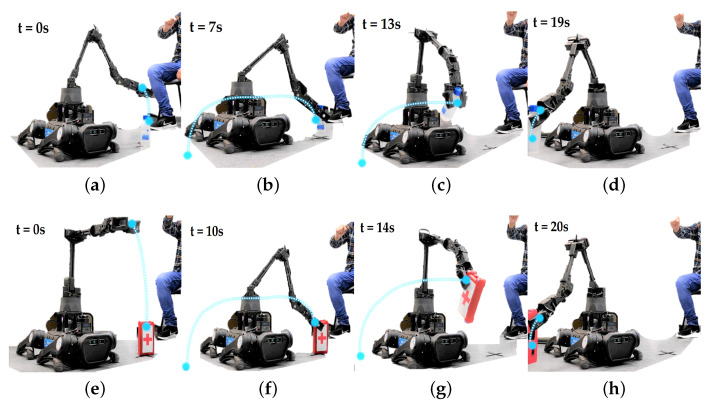
Manipulation tests for transporting objects (First kit aid (**a**–**d**) and Alcohol (**e**–**h**)), through MR Tele-operation. (**a**) Approaching the manipulator end-effector to the object. (**b**) Close the griper and capture the object. (**c**) Object transportation to the destination point. (**d**) Gripper opening to deposit the object. (**e**) Robotic arm in the home position, the destination point is in blue. (**f**) Close the griper and capture the object. (**g**) Object transportation to the destination point. (**h**) Gripper opening to deposit the object.

**Figure 7 sensors-22-08146-f007:**
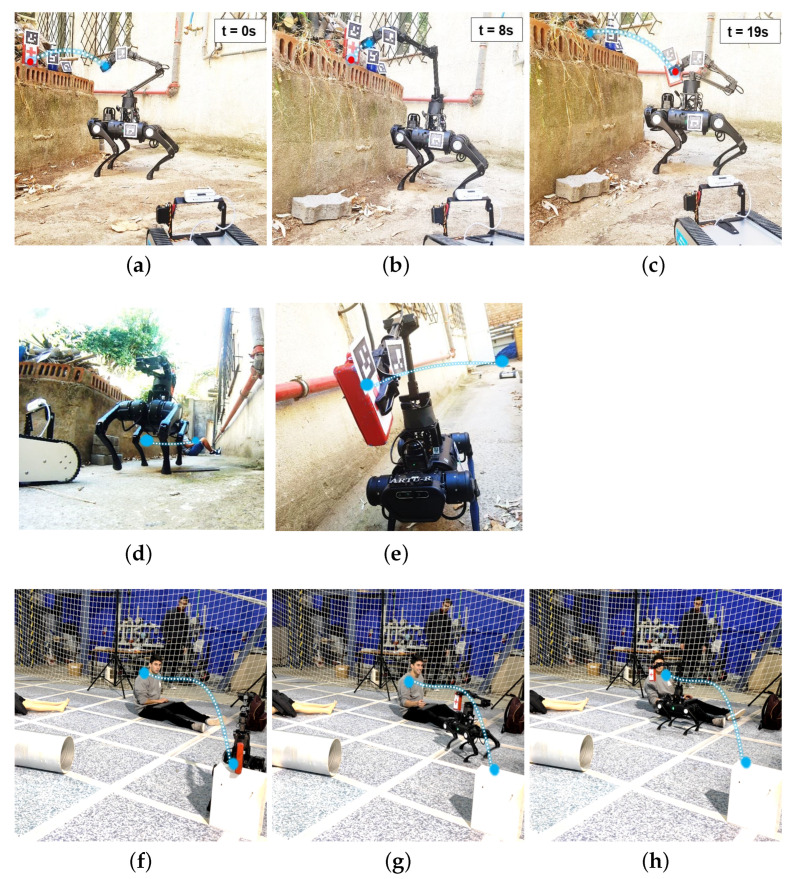
Outdoors: Test for manipulation payload in outdoors (**a**–**c**) using the MR system. Indoors: Test for robotic assistance transporting medical equipment to a victim (**d**–**f**). (**a**) First pose for payload manipulation. (**b**) Robot set grabbing payload. (**c**) Transport pose of the payload. (**d**) Robot transporting the payload to the victim. (**e**) Robot arriving the destination. (**f**) Robot tacking the first kit aid. (**g**) Robot transporting kit. (**h**) Kit delivery to the victim.

**Table 1 sensors-22-08146-t001:** Materials for the proposed system implementation.

Component	Description
Hololens (2nd generation)	Mixed-Reality Glasses
Hololens (1st generation)	Mixed-Reality Glasses
Unitree A1	Quadruped Robot
RP-Lidar	Lidar Sensor
WX250S	6DoF Robot Arm
Nvidia Jetson Xavier-Nx	On-board Embedded System
Real-Sense	RGB-Deph Sensor

**Table 2 sensors-22-08146-t002:** One leg Denavit-Hartenberg’s parameters. 3DOF.

Thetha (deg)	d (mm)	a (mm)	Alpha (deg)
q1 + 90	55	0	−90
q2	55	200	0
q3	0	215	0

**Table 3 sensors-22-08146-t003:** 6DOF Manipulator’s parameters.

Thetha (rad)	d (mm)	a (mm)	Alpha (rad)
q1	104.43	0	pi/2
q2 + pi/2 − 0.2	0	254.95	0
q3 + 0.2	0	0	pi/2
q4	250	0	−pi/2
q5	0	0	pi/2
q6	131	0	0

**Table 4 sensors-22-08146-t004:** Comparison of related works in the state-of-the-art.

Work	Kind of Base	Glasses	Simultaneous Control Real-Virtual
[13]	Legged Robot	HTC Vice	YES
[37]	Mobile Robot	Hololens 1	Visualization Only
[38]	Fixed	Hololens 1	NO
[39]	Fixed	Hololens 1	NO
[40]	Fixed	Hololens 1	NO
[41]	Wheeled platform	HTC Vive	NO
Proposed Method	Legged Robot	Hololens 1 and 2	YES

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
