# Peer review of "A Mixed-Reality Tele-Operation Method for High-Level Control of a Legged-Manipulator Robotâ€"

_sensors, 2022, doi:10.3390/s22218146_

Round 1
Reviewer 1 Report
Comments and Suggestions for Authors
The proposed paper describes an interesting approach to a Mixed-Reality Tele-operation Method for High-Level Control of a Legged-Manipulator Robot. In the last decade, quadruped robots have been the object of study within the field of robotics.
In the article, the authors presented a manipulator with six degrees of freedom mounted on a four-legged robot. They used Matlab and Gazebo environments to analyze the issue under consideration.
The main topic of the article is the implementation of the Tele-operation method for the ARTU-R and the 6 DoF robotic arm using Mixed-Reality (MR) technology. Such a solution increases the awareness of the operator's surroundings through a virtual model. This model via Hololens is added to the real world. Tests of the proposed control method were carried out according to prepared scenarios (taking into account National Institute of Standards and Technology specifications).
However, I have a few questions and suggestions:
1. Lines 72-73 - There should not be a new paragraph.
2. Figure 1 – Figure has no reference in the text.
3. Table 1 – The middle column is not needed because all values are equal to one.
4. Lines 105-106 – The authors should include mathematical formulas in the article to show the relations discussed.
5. Figure 4 – Why the authors use two types of Cartesian coordinate systems (right-handed and left-handed).
6. Authors should standardize abbreviations and descriptions throughout the article for example: MR, M-R – lines 176, 180.
7. Figure 6 – In my opinion, the descriptions under the drawings are not appropriate - please verify.
8. Figure 8 - The middle part of the picture with the computer and hololens goggles should be removed or moved elsewhere in the article. In its current location it creates confusion.
The research is poorly described; the authors should add a description of the surveys conducted with the operators. More detail about what the surveys were about. Questions, rating scale, etc. because looking at the charts posted supposedly something is better than the standard version, but I don't quite know why.
In related work, the authors could also refer to other novel methods of manipulator control (for example, using gestures or voice commands). Such examples can easily be found especially in the case of industrial robots:
· Rosenstrauch, M.J.; Pannen, T.J.; Krüger, J. Human robot collaboration—Using Kinect v2 for ISO/TS 15066 speed and separation monitoring. Procedia CIRP 2018, 76, 183–186.
· Kaczmarek, W.; Lotys, B.; Borys, S.; Laskowski, D.; Lubkowski, P. Controlling an Industrial Robot Using a Graphic Tablet in Offline and Online Mode. Sensors 2021, 21, 2439. doi: 10.3390/s21072439
· Kaczmarek, W.; Panasiuk, J.; Borys, S.; Banach, P. Industrial Robot Control by Means of Gestures and Voice Commands in Off-Line and On-Line Mode. Sensors 2020, 20, 6358. doi: 10.3390/s20216358
· Torres, S.H.M.; Kern, M.J. 7 DOF industrial robot controlled by hand gestures using microsoft Kinect v2. In Proceedings of the 2017 IEEE 3rd Colombian Conference on Automatic Control (CCAC), Cartagena, Colombia, 18–20 October 2017; pp. 1–6.
· Park, C.B.; Lee, S.W. Real-time 3D pointing gesture recognition for mobile robots with cascade HMM and particle filter. Image Vis. Comput. 2011, 29, 51–63.
The conclusions are very general. They should be expanded and clearly indicate the results obtained.

Author Response
Dear reviewer, in the attached document you can find the answers to your correct comments.
We greatly appreciate the suggestions that have allowed us to improve the quality of our article.

Reviewer 2 Report
Based upon my careful observation, the work and the presented results are good. Therefore, it can be published subject to the following modifications/corrections/queries:
1-The English writing of the paper is required to be revisited. Please check the manuscript carefully for typos and grammatical errors.
2-Please explain this control system of this robotic set, is it novel, otherwise please put necessary references. The authors are requested to add more details regarding their original contributions in this manuscript.
3-What is the merit of the current manuscript? In other words, what is the advantage of this proposed system? This is not clear enough. Please clarify to the reader.
4- What is the robustness/effectiveness the proposed system over the existing similar systems? Comparison should be done with published papers related to this thematic.
5- Future research direction should be shown in conclusion.
Author Response

(The authors gave the same response as above.)

Round 2
Reviewer 1 Report
Comments and Suggestions for Authors
The authors addressed my comments and made corrections to the article. However, the writing style in the article should be improved, e.g.:
1. Line 51 – Experiments carried out have been to (…) – have been carried out
2. Line 64 – section 4 (…)– Section 4
3. Line 164 – language (…) – language
4. Line 193 – buttons that allow execute (…) – buttons that execute
5. Line 204 – capable of adopting adopt (…)– capable of adopting
6. Line 220 – 10 participants in the roll (…) – in the role
7. Line 227 – who used (…) – using
8. Line 241 – take a medical (…) – taking
9. Line 243-246 – sentence must be changed
Figure 4-b - all variables in the diagram should be explained below.
Figure 8:
· Speed to complete task (m/s?) – Time to complete task ? (also in line 280)
· Portabilidad – Portability ?
· Easy to use – easy of use ? (also in line 280)

Author Response
Dear reviewer, we appreciate your time to review our article, following your observations, minor revisions have been addressed. Attached is the document with the changes made to facilitate this review phase.
